# University central offices are moving away from doing towards facilitating science communication: A European cross-comparison

**Marta Entradas**[1,2]*, **Frank Marcinkowski**[3], **Martin W. Bauer**[2], **Giuseppe Pellegrini**[4]

**1** Department of Sociology, Iscte-Instituto Universitário de Lisboa (Iscte-IUL), Lisbon, Portugal,
**2** Department of Social Sciences, Heinrich-Heine-Universität Düsseldorf, Düsseldorf, Germany,
**3** Department of Psychological and Behavioural Science, London School of Economics and Political Science, London, United Kingdom, **4** Dipartimento di Sociologia e Ricerca Sociale, Università degli Studi di Trento, Trento, Italy

* marta.entradas@iscte-iul.pt

**Data Availability Statement:** The SPSS file is available from the flagshare database (DOI: 10.6084/m9.figshare.23826057).

## Abstract

There is increasing interest in studying science communication from an institutional point of view. With much of the empirical research focusing on views of institutional actors on communication and their roles in the organisation, less attention has been paid to practices and dispositions of universities to communicate their research with publics. Universities have professionalised communication structures for external relations, and science communication has been absorbed in this. Yet, the evidence on what those practices represent for the university—at different levels of the organisation—is insufficient to understand the role of science communication within the university landscape. This study investigates science communication at central offices of research universities. Sampling whole populations of universities in four European countries (Germany, Italy, Portugal, and the United Kingdom; 44% response rate), we disentangle practices of communication as a centralised function. To the best of our knowledge, this is the first cross-national study on this topic based on all universities within the surveyed countries. We compare general trends in science communication of universities across countries. The evidence shows that science communication is a secondary function at central offices of universities, strongly medialised, and points to a supporting role for central structures in facilitating science communication at other levels while moving away from doing it themselves. Universities might need to consider their long-term positioning in enhancing national science culture by fostering science communication through models of dialogue and public debate.

## Introduction

There has been a visible trend towards embedding science communication in universities and research organisations internationally [1, 2]. While many factors can be attributed to this, one that is distinguishable is the recognised role of the university in contributing to the common good and societal needs [3] by stimulating the social 'conversation' about science and engaging

**Funding:** This research received funding from Fundação para a Ciência e Tecnologia (FCT), with Grant agreement No. PTDC/COM-OUT/30022/2017. The funders had no role in study design, data collection and analysis, decision to publish, or preparation of the manuscript.

**Competing interests:** The authors declare no competing interests.

publics [4]. In modern research, as the contextualisation of research in society has changed—science should serve society and society should participate in it—ideals of communicating science have also evolved towards engaging the broader society in discussions and scientific and technological developments and innovations that respond to their needs. This goes beyond the traditional media relations of universities [5] and requires new communication models such as public events and fora for public engagement. Although there will probably always be an 'epistemic dependence' [6] of the public on others' knowledge, there has emerged a clear concern at both national and European policy levels [3] to create opportunities and formats where scientists and non-scientists can discuss face-to-face, openly, without expert interdependence and hierarchy [7–9].

At least rhetorically, science communication has become part of the university discourse. Communication of science and public engagement are seen under the umbrella of the 'third mission' of universities as important to establishing relationships with various stakeholders, industry, and the public more broadly [10, 11]. Yet, their efforts for communicating research have often been associated with a promotional culture of the university [12], and intended medialisation of science [13] to search for broader visibility and self-promotion rather than public enlightenment or engagement [14, 15] as they face multiple competitions (students, staff and funding) [16–18]. Scholars have pointed to challenges for science communication and public engagement to thrive in universities, arising from 'capitalist pressures' in science, marketisation and commercialisation of research, which 'sells science' [19] through PR and marketing research and transforms science communication in propaganda [20].

As much of university communication might be absorbed by PR activities and such university priorities, the empirical research is still scarce on the role of the *science communication function* [21] within these organisations. One of the challenges in investigating science communication practice is the difficulty in distinguishing what it is from what it is not. Entradas and co-authors (2023) disentangle theoretically and empirically the science communication function from other corporate functions such as PR, Marketing (MA) and Public Affairs (PA), which have also been invoked in public communication of research [22]. Another challenge is the complexity of the modern university, its different hierarchies and missions, and its various organisational levels where communication of science can take place. Is the focus on the centralised approach of the university central communication office (central level), the decentralised activity of research institutes and departments (meso-level) [21], or even individual scientists' public communication (individual level) who speak in the name of the university [23]. As such, the study of the role of science communication in the organisational environment requires a broader perspective that considers the various organisational levels concerned with the management of university public communication and image.

Research that has started to examine communication activities at the meso-level finds evidence of expanding science communication practices of research centres, institutes, and departments, with public events, traditional media and online contacts led and resourced locally [24, 25]. These efforts are less media oriented and more directly public focused, and are less motivated by self-interested rationales [21, 26], thus suggesting a growing commitment to science communication at this level. The authors further argue that this might contribute to making research institutes potential niches for science communication with more localised practices and events, while more selective, 'visible' science becomes communicated at other levels (too). Thus, the decentralisation hypothesis [21, 27] claims that science PR and marketing takes place at central communication structures while science communication of the more disinterested kind becomes an activity at decentral offices. This does not mean however that no activities can be shared across levels and occur in collaboration, but rather there is a visible pattern of communication at different levels. Yet, to investigate and give meaning to this

hypothesis, we must learn about the broader spectrum of activities of the central communications offices to communicate science. We will investigate types of activities that a university might use to communicate about and/or engage a public in research, including public events, traditional media, and new media; these activities might take various formats ranging from the traditional one-way dissemination of public lectures and media communication to two-way communication that can involve dialogue and participation in research and decision-making. And we examine the conditions (infrastructure) in which this activity is integrated in the dynamics of central communication offices by looking at resources and rationales driving these efforts.

## Research questions and hypotheses

In this study, we ask (RQ1) 'what *communication activities* do central communications offices organise to communicate about their research, to which *audiences*, and what *rationales* characterise their efforts'. From this main research question, and based on the existing literature, albeit limited, we develop our hypotheses, and RQ2.

Studies in the PUS literature that shed light into the organisational embedding of science communication at university central communication structures have mainly focused on scientists' interactions with Press and PR offices (and officers) [28, 29], views and roles of communication professionals [30–34]. From these studies, perhaps not surprisingly, we learn that these structures have traditionally communicated research through media outlets, with communication professionals' tasks being mainly the production of press releases and responding to journalistic enquires while also playing an important role in supporting scientists in their media communication and visibility. This tendency for institutional science PR is not new. It had already been noted in Nelkin's account of science in the media at scientific institutions [19]. Yet it has only been boosted over the past years as seen by the increase of university press releases (e.g. [35, 36]) or media centres as reported by Japanese universities as essential to reach the national media [37]. At the same time, new media including social media platforms like Facebook, Instagram, Twitter, and YouTube, has centralised as a strategy for institutional communication [38] also for communicating science (e.g. [39, 40]); however, their capacity for societal engagement has been found limited [41]. Although the evidence points to strong media activity, these are localised studies with a focus on media communication, and we still lack an understanding of how this potentially dominant media activity compares with other communication formats that universities use to communicate and share research with external audiences. This leads to our first hypothesis:

H1a: *Central communication offices put higher efforts on media activities for communicating science, both traditional and online, than on public face-to-face events,*

H1b: *For central communication offices media audiences are preferred targets over general publics.*

This attention seeking through the media is seen also at the management level of the university as a priority, with consequences for central communication offices that implement managerial views, while also providing indication on the motivations of universities. For example, a study with decision makers of universities in Germany (executive management board members) shows that they attribute significant importance to mass media coverage and visibility of their research. In mediatised universities, PR managers are given more power and are more likely perceived as 'public experts' [31], although overall PR managers are not given (yet) much importance by management. Also, communications staff working in UK universities have highlighted that public engagement is perceived within universities management as

servicing more instrumental goals specific to research governance while lacking support in the organisation (e.g. funding, trained staff) [42, 43]. This is despite communication professionals' recognition of the role of the university in contributing to civic engagement and public debate about science and technology [30, 44]. This suggests that the 'media logic' and instrumental goals are in fact interlaced with the institutional communication logic in universities at the management and centralised level, for wider and also science communication. A propagandistic approach would likely be driven by the need to disseminate the 'ideology of academic entrepreneurship' and guided by self-interested rationales (e.g. recruit students, raise profile, get public support) rather than rationales oriented towards a 'societal conversation' (e.g. disseminate research, stimulate public debate). This leads us to our second hypothesis:

*H2*: *Universities' central communications are more likely to recognise 'self-interested' rationales than 'public-good' oriented rationales.*

Additionally, we examine whether there are divergent or convergent patterns in communication practices across the four countries studied. We do not have good reasons to expect differences in the tendencies of university communication at central offices and 'national ways' of communication. A common ecology of universities in a tertiary education market rather suggests a convergence of response across institutions seeking to solve similar problems. A tendency for 'European unification' in the external presentation of universities follows EU policy towards a single higher education 'market' [45] and in this light the adoption of third mission ideals. If anything, we would expect specifics for the UK universities, which in the post-Brexit environment might lead to even stronger marketisation than continental Europe, though it might be early to see any such effects. Thus, it seems plausible to expect that the practices of communication in universities respond to a similar set of problems and their solutions will therefore be analogous showing little variation between these four countries. We pose the hypothesis below, and we describe these along with the examination of H1 and H2.

*H3*: *There are convergent similarities in science communication (activities, audiences, and rationales) at central communication offices across countries.*

Finally, we were interested in understanding factors that can predict science communication activity at this level. We consider the effects of contextual (C) factors, which refer to features of the university and research environment (e.g. country, size, research funding), and disposition factors (D factors), which reflect the conditions and formal commitment (infrastructure) that support the development of such activities (e.g. institutional policy/guidelines for public communication, professional communications staff, engagement of scientists in institutional initiatives). Previous research shows the importance of these factors for science communication at the meso level [46]; we test the separate and combined influence of these factor classes at the central level as well, and we also explore the influence of 'rationales' on institutional public communication, one of the examined dimensions in this study. It would be reasonable to expect that a university that recognises the public value for science and a role in facilitating public engagement in it, would tend to increase their resources and communication efforts targeted at general publics. We ask:

RQ2: What factors predict public communication activity at central communication offices?

## Methods

### Data and data collection

The data is part of the international research project "OPEN: Organisational Public Engagement with Science" (2019–2023), which investigated public communication in university

settings with a focus on science communication, with data collection in Portugal, Italy, Germany and the United Kingdom. In all countries, we conducted a whole population survey of public and private universities. The exception was for the UK that has a broader population; in this case, we listed all universities that were part of the REF (Research Evaluation Framework, 2018, N = 154), i.e. the receivers of funding from major UK government budgets.

The national surveys were distributed online between May 2020 and March 2021 and the data was centrally collected in Lisbon by using the Qualtrics platform. This included two phases of data collection and follow-up phone calls to encourage further participation. All correspondence with universities in the four countries was in the local languages of German, Italian, and English. The questionnaire was designed in English, translated into local languages, and back translated into English by the national teams. In Portuguese universities the study was conducted in English as this is the common academic language in use. The questionnaire enquired about practices, motives, and communication dynamics across levels of the university with a focus on central communication offices.

## Sample design, sample of target respondents

We aimed at sampling *the* central communication offices of universities, i.e the main communication office of the university. In most cases there is one under different names; Comms Office, Comms Department, or PR office are some of the most common names we encountered. When a PR, Marketing or Press office existed as a separate office (as in larger universities), we always aimed at the general, central communications office, where we expected a variety of functions to cohabit. Each national team listed all research universities in the country, i.e. universities involved in scientific research and which apply for national funds. These lists were either provided by national governments (in the case of Italy and Portugal) or built from scratch with information from official websites (such as the Ministries of Education and/or Research, government funding agencies, universities websites, etc.). For Germany, we used the website of 'HRK-The Germans' Rectors Conference: Voice of the universities', and for the UK, we used the REF website (2018). For each university, we collected general information such as the name, URL, postal address, location, year of foundation, size (students enrolled), type of institution, type of funding (private, public), right to doctorate (yes/no), level of excellence, and collected information about the central communication office including the contact point (name, position, postal address, email, telephone). The latter was our target person for this survey, someone who could speak for the university strategy and practices at the central communication office, and was a professional responsible for overseeing communication (director of communication, project managers or officers).

No personal data about these communication professionals were collected; the online survey focused on institutional practices. Electronic consent was requested for participation: participants were informed at the beginning of the survey about the nature of the study, that participation was voluntary, no risks were foreseen, and confidentiality was assured (responses were anonymous with no possibility of identifying the participating institution or participants).

We contacted $n$ = 719 universities in the four countries and received $n$ = 319 responses. This accounts for an overall response rate of 44% (see Table 1 for breakdown per country). Most respondents identified themselves as communications staff (62%), management staff (21%), administrative staff (2%), academic staff (6%), or others (5%).

## Measures

To investigate the posed research questions (RQ) and hypotheses (Hs), we use the responses to the questions described below.

**Table 1. Number of universities contacted, universities that responded, and response rates, by country.**

| | Universities contacted | | Universities responded | | Response Rate (%) |
|---|---|---|---|---|---|
| **Country** | (N) | (%) | (N) | (%) | |
| Germany | 368 | 51 | 124 | 39 | 34% |
| Italy | 97 | 13 | 92 | 29 | 95% |
| Portugal | 100 | 14 | 63 | 20 | 63% |
| United Kingdom | 154 | 21 | 40 | 13 | 26% |
| **Total** | **719** | **100** | **319** | **100** | **44%** |

*Surveying the central level.* To study the central level, it is crucial to record *only* 'central level activities'; surveying central communications offices does not necessarily mean measuring only central activities or science communication activities. This requires a special questioning technique, which we have implemented. We explicitly asked the communications responsible (head/director/manager) at *the* central office of the university—one office per university—about what central officers do themselves, what gets done elsewhere but is supported by them, and what is done without their own contributions at other levels, and importantly, what activities are intended to communicate about research. This approach is different from previous studies that regularly report on all the activities they know about, regardless of whom initiated and organised them, i.e. regardless of whether it is a central activity or not.

**Dependent variables.** The blocks of questions on communication activities introduced the statement "In the following questions, we are interested to know what your university's preferred means are to communicate about science and research", and repeated in the single questions. And, we added a single question enquiring directly, "how much of the overall communication at the central offices is for communicating science?"

*Public events and traditional media channels (14 items).* To examine the occurrence of different activities, we asked on a scale from (1) never, (2) annually, (3) Quarterly, (4) Monthly, (5) Weekly, and (6) Don´t know "Roughly, how many times in the past 12 months have you used the following to disseminate science contents? Please refer only to activities about science and research": (1) public lectures, (2) public exhibitions, (3) open days/guided visits and similar events, (4) science festivals/fairs, (5) national science week and similar events, (6) science cafes and similar formats of public discussions, (7) citizen science projects, (8) policy making events, (9) events with private institutions (industry/business), (10) interviews for the media, (11) press conferences, (12) press releases, (13) articles in magazines/newspapers, (14) brochures/leaflets/other university publications.

*New media channels (5 items).* We asked "Roughly, how many times in the past 12 months has your communications office used the following new media to engage with external audiences? Please refer only to activities about science and research": (15) institutional website, (16) blogs, (17) Facebook, (18) Twitter, (19) podcasts/multimedia. The intensity of use was measured with a five-point scale from (1) Never, (2) Quarterly, (3) Monthly, (4) Weekly, (5) daily, providing also a (6) Don't know option.

In addition, we explored media interactions with the questions: "Do you maintain a list of journalists?", for options: (1) yes, we have a list of journalists; (2) no, but we have personal contacts; (3) no, we do not have a list/database of journalists. And "When journalists want to contact researchers at your university how do they proceed?", with options (1) They contact the central office first and we put them in contact with the researchers, (2) sometimes they contact us, other times they contact the researchers, (3) they often contact the researchers/departments/schools directly.

*Estimated number of activities and indices of intensity*. We estimated the number of activities by recoding the above variables in number of participations, and assuming 48 work weeks per year. Coding was as follows: never (= 0), annually (= 1), quarterly (= 4), monthly (= 12), weekly (= 48), and daily (= 240).

From these variables, we built indices of intensity of public communication activity for the three communication means: the 'public events' index refers to the sum of participations in activities (1) to (9), the 'traditional media' index is given by the sum of items (10) to (14), and 'new media' sums up the number of interactions in items (15) to (19). Reliability analyses show high internal consistency for the 3 indices (Cronbach's α = .734 for public events, .674 for traditional media, and .785 for new media) (S1 Table).

**Independent variables.** *Country*. Country is a categorical variable where (1) is Germany, (2) Italy, (3) Portugal, and (4) United Kingdom.

*Size of university*. This is a continuous variable referring to the number of students enrolled in the university (M = 11536; SD = 16734; Median = 5038). We note differences in the size of universities per country (p < .001), with universities in the UK (M = 12.508, SD = 11.879, Median = 8500), and Italy (Median = 12.500) being larger than universities in Germany and Portugal (Median = 4036 and Median = 3100, respectively).

*Size of communications staff*. We asked "How many people work in the central communications office? Please consider all employees who carry out communication tasks as part of their day-to-day responsibilities. This can include people responsible for maintaining the website, social media, organising public events, producing the newsletter, responding to journalists, etc. For simplicity we will refer to them as communications staff".

*Size of science communication staff*. And, we asked specifically about communications staff dedicated to science communication tasks: "how many of the 'communications staff' working in the central communications office have science communication tasks as their job description, i.e. dedicate time to tasks related to communicating about science and research. Please indicate the number". These are numerical variables, strongly correlated; as such in the regression analysis we use only the overall number of communication staff (that also has a larger *n* of responses).

*Funding for communication activities*. We asked "Please estimate the percentage of the university budget spent on public communications in the last 12 months. This can include actions such as maintenance of the website, printing of brochures, organisation of public events, etc. Please do not consider salaries of the communication staff". Options were < 1% (coded 1), 1–5% (coded 2), 6–10% (coded 3), > 10% (4), Don't Know (5). This variable was recoded into a binary for the regression analysis where (0) is 'low' (less than 1%) and (1) is 'high' (1% or more).

*Policies for public communication*. We asked "Please tell us whether the following statements about your university commitment to public communication are true or false: (1) 'our university has a public communications strategy', (2) 'our university has a policy encouraging public communication of science and research', and (3) 'our communication efforts respond to national policies for societal engagement in science'. All countries in this study have, in one way or another, some sort of policy/guidelines, for instance, in Portugal there is a national policy for scientific culture, and in the UK, there are reports that have become policy marks internationally. This was a (1) true, (2) false question with a (3) Don't know option. For the regression analysis, we used only item (2) 'science communication policy' as an indicator of a formal commitment to science communication, and recoded it into a binary variable 'no' (= 0) and (= 1) 'yes'.

*Active researchers (participation in SC)*. We asked about the involvement of scientists in the activities organised by central communication offices. This was found to be an important

factor for the activity of research institutes, which is higher when more researchers respond to the requests of the communications staff (25). Here we test the predictive power of this factor by asking "Roughly, in the last 12 months, what percentage of the academic staff and researchers in your university took part in public communication activities you organised such as public lectures, public debates, media enquiries, etc?". Options were (1) 1 < 10%, (2) 10–20%, (3), 20–40%, (4) 40–60%, (5) 60–100%, and Don't know (6). (M = 2.8, SD = 1.5, Median = 3). This variable was recoded into a binary for the regression analysis where (0) is 20–40% or less, (1) is more than 20–40%.

*Target audiences*. We asked "In the table below you will find a list of external audiences you might contact and direct your communications. How often have your communications efforts been addressed to each of the below in the past 12 months?". This question was measured on a three-point scale from never (= 1) to (= 3) frequently, and a 'Don't know' option was provided. Audiences were: (1) general public (whoever might be interested), (2) schools, (3) prospective students, (4) members of local municipalities/councils/associations, (5) delegates from industry, (6) governments/politicians/policymakers, (7) non-governmental organisations (NGOs), (8) media and journalists, and (9) (peers/other universities/people like us). We do not include these variables in the regressions because they are not explicative variables in our model; the focus is captured in other questions (e.g. on rationales) and the variance of these audiences is already reflected in other predictors.

*Rationales for communication*. Respondents were asked to rate by level of importance fourteen items, reflecting concepts of interest for the institution and for the 'other'. Coding (= 1) is very important, and (= 5) is not important. Items were: (1) to respond to the university mission, (2) to respond to requirements from funding bodies, (3) to respond to national policies for public engagement with science, (4) to raise the university profile, (5) to attract funding, (6) to give back to taxpayers, (7) to get public support for the university, (8) to disseminate the university research to the public, (9) to recruit students, (10) to listen and involve the public in the university activities, (11) to stimulate public debate, (12) to influence policy, (13) to contribute to the legitimacy of science in the public sphere, (14) to further the career of the university staff (see S2A Table for descriptive of items).

Factor analysis using principal component analysis on all fourteen 'rationales' show that items loaded appropriately in two factors, confirming a two-dimensional structure of the construct rationales (KMO = .89, p < .001, 51% of variance explained); the loadings are shown in S2B Table. We labelled Factor 1 *'public-good'* oriented referring to the perceived rationales of giving back to taxpayers, disseminate research, engage the public or contribute to the legitimacy of science, that is, a rational of "affecting the public" reflecting a perceived benefit to the other; and Factor 2 *'self-interested'* referring to perceived benefit to the institution; this factor aggregates items such as 'attract funding', 'recruit students' and 'raise the university profile'. Reliability tests show high internal consistency for 'public-good' oriented (Cronbach = .87) and self-interested (Cronbach = .71). A higher value represents stronger 'public-good' oriented and self oriented rationales (i.e., agreement that it is important to communicate for the institution or for the public). These indices were recoded into binaries using a median split for purposes of descriptive analysis, but we use the factors scores in the regression analyses.

## Analyses

We use descriptive analysis to report characteristics of central communication offices, and address our main research question and its related H1 (activities and audiences) and H2 (rationales). For country comparison (H3), we estimated one-way ANOVA models and pairwise Bonferroni post-hoc tests by country to determine which countries differed significantly from

each other in the activities organised, the main audiences addressed, and the rationales guiding them; these comparisons are described along H1 and H2. RQ2 is addressed with three-step hierarchical regression models that investigate the effects of the independent variables (C and D variables) on the level of public communication activity (three separate models for each independent variable 'public events', 'traditional media', 'new media'). Step 1 (Models 1), introduce *size* and *country*, step 2 (Models 2), add *communications staff*, *funding*, *policy/guidelines*, and *active researchers*, and step 3 (Models 3), bring *rationales*. We report Unstandardised Beta (B), and consider $p < .05$ for significance.

## Results

### Central communication offices

Firstly, we briefly describe general characteristics of the central communications offices, i.e. the infrastructure in place, such as the size of communications teams and time they allocate to science communication tasks, the policy/guidelines and the funding available for communications, and we compare these across countries.

**Communications staff.** Overall, the data point to well established central communication structures at universities in all the countries studied. Central communication offices employ on average eight communication professionals (M = 8.0, SD = 9.6, who have been in their current positions for about 9 years (M = 9.2, SD = 7.9, Median = 6), and this number has increased in about half of the universities (56%) in the past five years. Of these, less than three persons dedicate time to science communication tasks (M = 2.8; SD = 3.4). The others are mainly involved in PR and marketing activities [22]. This alone is indicative of the subordinate role of science communication at the central level.

Most communications staff have a background in communications, either at the undergraduate (M = 3.2, SD = 28) or postgraduate level (M = 3.1, SD = 28), and about two persons have training in science communication (M = 2.1, SD = 28), still a large number has been trained on the job (M = 4.9, SD = 28.8). We note that communication teams are significantly larger in UK universities. For example, in the UK there are about four persons dedicated to science communication tasks contrasting with 1.4 persons in Portuguese universities. This could be in part related to the size of the universities, as suggested by the positive correlations between size, and 'staff', but there might be country commitment to public engagement as well as we discuss further.

**Funding for public communication.** As for funding allocated to communication activities, the average university spends less than one percent of their overall budget on communications carried out at central offices, a percentage that excludes staff salaries. This however might not represent the overall spending on university communication, as for instance, marketing and advertising activities are distributed in different ways and structures across the university and the amount might not be known to these professionals—the low response rate to this question is in itself evidence of this. There is in fact evidence that most marketing expenditure come from outside the central offices. According to Educational Marketing Group (2019), public and private nonprofit institutions typically spend anywhere from 1.5–6.0 percent of the institution's annual operating budget on marketing. Although the spending on communications represents only a small proportion of the universities' annual budget, this investment has increased significantly over the past years, but this investment is lower when compared to other sectors (e.g. companies in the service sector invest as much as 8–10% in marketing and advertising).

About half of universities (52%) (total respondents n = 188) spend less than 1% of the overall university budget, about 38% spend between 1% and 5%, while only about 5% of the

**Table 2. Characteristics of central communications offices compared across countries.** The table summarises reported SC activities (events, traditional and new media channels), comms staff, funding, policies, and rationales by universities in each country ($n$ = 319).

| | Germany | | Italy | | Portugal | | UK | | Total | | |
|---|---|---|---|---|---|---|---|---|---|---|---|
| | Mean | SD | Mean | SD | Mean | SD | Mean | SD | Mean | SD | p value |
| *SC activities* | | | | | | | | | | | |
| Public events (estimated number of face-to-face events) (n = 319) | 45 | 51 | 60 | 56 | 51 | 44 | 49 | 52 | 51 | 52 | 0.25 |
| Traditional media channels (estimated number of contacts) (n = 319) | 58 | 38 | 110 | 64 | 67 | 51 | 117 | 55 | 84 | 57 | .000 |
| New media channels (estimated number of posts) (n = 319) | 340 | 298 | 542 | 360 | 371 | 311 | 451 | 355 | 423 | 338 | .000 |
| *Communications staff* | | | | | | | | | | | |
| Communications staff (number) (n = 316) | 5.6 | 5.0 | 7.2 | 7.3 | 5.2 | 4.6 | 20.0 | 18.0 | 7.8 | 9.6 | <0.001 |
| Comms staff dedicated to SC (number) (n = 312) | 2.2 | 3.0 | 3.3 | 4.3 | 1.4 | 1.1 | 4.1 | 3.4 | 2.6 | 3.4 | <0.001 |
| *Communications funding* | | | | | | | | | | | |
| Comms funding (> = 1%) | 51% | | 47% | | 50% | | 37% | | 47% | | p = .672 |
| *Communications policy* | | | | | | | | | | | |
| 'We have a communications strategy' (yes) (n = 308) | 65% | | 87% | | 89% | | 66% | | 76% | | .000 |
| 'We have a SC policy/guidelines' (yes) (n = 307) | 68% | | 92% | | 91% | | 86% | | 82% | | .000 |
| 'We respond to national SC policies/guidelines' (n = 305) | 70% | | 92% | | 88% | | 85% | | 82% | | .000 |
| *Rationales for communication** | | | | | | | | | | | |
| Self-interested (% high/low) (n = 285) | 47% | | 48% | | 52% | | 60% | | 50% | | 0.51 |
| Public-good oriented (% high/low) (n = 285) | 33% | | 66% | | 64% | | 45% | | 51% | | .000 |

* Percentages for high (percentage for low is not shown but it sums to 100%).

universities spend more than 5%. We note however that a significant number of responses to this question were 'Don't Know' answers (n = 113; 35%) suggesting difficulty in responding to this question indicating some distance of these professionals from management decisions; in many universities, budgets concern university leadership and management in which these professionals might not get involved. This is comparatively low if we compare with other sectors. This pattern of spending is similar in all surveyed countries.

**Policies for public communications.** As for policies in place for public communication, about 76% reported having a public communications strategy, 82% have a 'policy encouraging public communication of research', and 82% say that their 'communication efforts respond to national policies for societal engagement in science'. German universities are however less likely to report communication policies than Portuguese and Italian universities (p < .05). All descriptive findings are reported in Table 2.

## Public events, traditional media, and new media channels for science communication

Traditional and new media channels are overall more frequently used by the central offices to communicate about research than face-to-face events, confirming our H1. The most frequently reported *public events* were public lectures (M = 14; SD = 17.4, 89% of the universities reported them), open days (M = 6, SD = 10.8, 90%) and public exhibitions (M = 4, SD = 7.9, 77%). These activities seem to indicate support from central offices towards science communication activities rather than full control and suggest collaboration with others levels—institutes and researchers—a public lecture or a public exhibition need researchers but it also needs the infrastructure to take place.

On the contrary, fewer universities engage in citizen science activities (M = 4; SD = 9.3; 58% reported them), science cafes and discussions (M = 4; SD = 8.0; 57%), events with private

institutions and industry (M = 4; SD = 5.7; 71%) and with policy makers (M = 2.0; SD = 3.0; 56% reported them). These activities have been found to be more commonly carried out by research institutes [25].

The most common media channels were press releases and media interviews, with universities reporting an average of one press release per week (M = 48; SD = 20.2; 96% of the universities write them), and one media interview (for the radio, newspapers and TV) every two weeks (M = 22; SD = 19.8; 96% give them), and about 70% reported organising press conferences, at an average of four per year (M = 4; SD = 7.5). University brochures are important for 90% of the universities (M = 10.0 per year; SD = 14.5), articles for magazines and newspapers are as well (M = 16; SD = 21.1). Also, when asked specifically about media relations, in only in 3% of the universities surveyed do the journalists contact the researchers/departments themselves; the norm is for journalists to contact the central offices, who then put them in touch with their researchers. Also, when researchers look for a media contact, in 90% of the universities it is the central office possessing a list of journalists' contacts, that facilitates the relationships.

*Online communication* is frequent, with the most popular online channel being the institutional website: half of the of the universities (49%) update it daily. Podcasts are a popular means too with about 85% of the communication offices producing an average of 29 pod episodes annually (M = 29; SD = 57.2). In addition, universities are strong users of social media networks: about half share information on Facebook daily (45%; and 96% of all universities use it); about 80% use Twitter, mostly on a daily (38%) or weekly basis (26%), and about 20% do not use it at all. Blog entries are not among the most popular, still about 55% published a blog entry in the year previous to the survey. We examine differences in the use of these means for communicating university research across countries below.

Overall, central communications offices that more actively organise face-to-face events show increased media contacts both through traditional and online channels, as suggested by the positive correlations between the indices.

**Science communication activities vary across countries.** One-way ANOVA models reveal significant differences in the use of traditional media channels (F(3, 286) = 25.333), p<0.001) and new media channels (F(3, 286) = 7.015, p<0.001) across the four countries, but no differences for public events (F(3, 286) = [1.374], p = 0.251) (see S3 Table). Bonferroni tests for multiple comparisons show that the mean value of traditional media channels was significantly higher in Italian and UK universities than in German and Portuguese universities. For example, Italian and British universities reported an average of 117 and 110 contacts, through traditional media (respectively), compared to 67 and 58 reported by Portuguese and German universities respectively; similarly, Italian and UK universities reported higher use of social media for communicating research contents, with Italian and British universities reporting on average 540 and 450 posts per year, respectively, compared to 380 and 340 reported by Portuguese and German and universities. The pattern changes when it comes to public events, where, despite non-significant results in the ANOVA, the pattern is different in universities in Portugal and Italy (see Fig 1- where a more coherent pattern is found in these countries, with visible higher activity in public events (Median = 42 and 38 events), compared to universities in Germany (Median = 28) and the United Kingdom (Median = 28); in Germany and the UK there are however a number of universities with high activity, adjusting the means, and possibly justifying the lack of significance found in the ANOVA (which compares means).

It is also in these countries, Italy and the UK, that we find larger variation among universities (as showed in the whiskers), meaning that some universities have a significantly higher media activity than others. This could however be an effect of the size of universities rather than a general characteristic of universities in these countries; we will control for this possible effect in the regressions. As for new media channels, there are differences between German

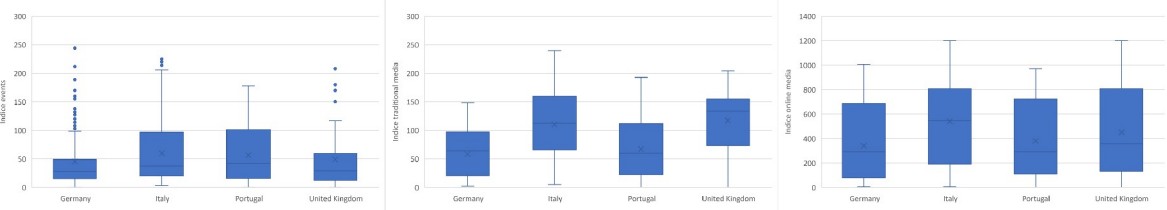

**Fig 1. Public events, traditional media, and new media channels contacts by universities in Germany, Italy, Portugal, and the United Kingdom (*n* = 319).** Charts represent the estimated number of participations in public events, number of media contacts in traditional means, and number of posts in new media channels.

and Italian universities (p<0.001), with central communication offices in German universities using less frequently digital media channels. Italian universities show the highest use of social media, and German the least. For example, Italian universities reported an average of 542 online updates/posts per year compared to 340 posts in German universities.

## Target audiences for university public communication

When asked about the frequency of contact with various public audiences, about 90% of the surveyed universities said they address frequently prospective students, general publics, and the media. Less frequently addressed were NGOs (13% addressed them frequently), members of industry (28%), government and policy actors (30%). This contradicts in part our expectations in H2 of a stronger focus on stakeholders and less on public audiences although it confirms that media audiences are indeed a main target.

These trends are common across countries, i.e. in general, the main target audiences remain the same in all the surveyed countries, yet there is variation in the intensity these audiences are addressed by universities in some of the countries. Bonferroni tests for variable 'students/public/media' and variable 'stakeholders' show only small differences: German universities that tend to address less frequently 'stakeholders' than universities in the other countries, 'public/students/media' are less addressed by Portuguese universities when compared to universities in other countries. These findings seem to suggest that overall, universities are competing for the same audiences, but it might be early to say where the tendency is going.

## 'Public-good' oriented and 'self-interested' rationales

From a list of fourteen communication motives, universities stated as most important 'to recruit students' (reported as very important by 72% of the universities), 'to raise the university profile' (63%), and 'to respond to the university mission' (61%). This contrasts with the lower importance attributed to rationales such as 'to contribute to the legitimacy of science', which was very important for 33% of universities, 'to stimulate public debate' (25%), 'to influence policies' (21%) or 'to give back to taxpayers' (16%), while already pointing to a stronger importance paid to self-oriented rationales than public-good oriented. These items, which were reduced to two factors, are represented in Fig 2. The figure shows that 'self-interested' rationales are important for most universities, regardless of the country, and 'public-oriented' rationales assume greater importance in universities in Portugal and Italy (medians are visibly higher).

## Motives vary across countries

When comparing rationales (two factors), we find differences between groups for 'public-good' oriented (F(3,255) = 12.61, p < .001) but not for self-interested rationales (F(3,255) =

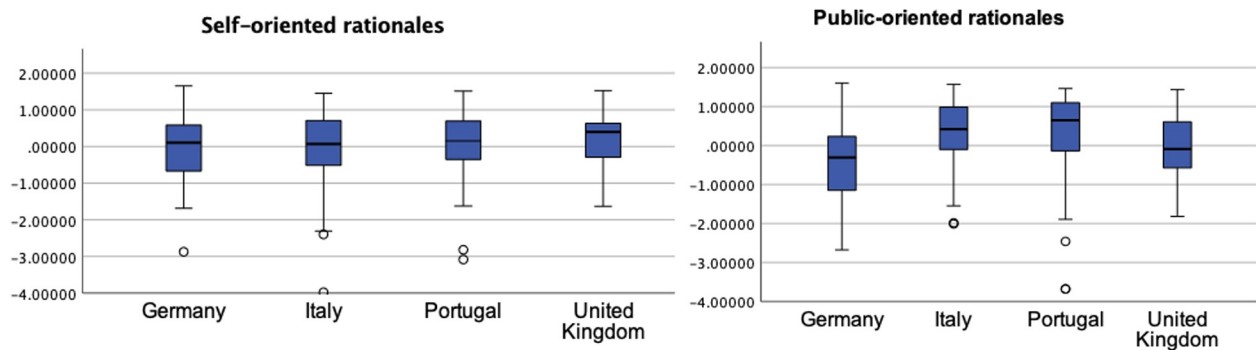

**Fig 2. Self-interested and 'public-good' oriented motives compared across universities in Italy, Portugal, Germany, and the United Kingdom (n = 319).**

.84; p = .475) (S4 Table). This might be explained by the fact that 'self-oriented' rationales are seen as highly important by most universities in all countries. Public oriented rationales gain in importance among universities in Italy and Portugal, not so much in universities in Germany and the UK that are less oriented towards engaging publics. Post hoc comparisons indicate that the mean score for the 'public-good oriented' motives in German and British universities was significantly lower than in Italian and Portuguese universities (p < .001), suggesting a lower interest in the values of public engagement among German and British universities.

The data shows in Fig 2. that the importance of instrumental communication is beyond question in all countries. The majority of values obtained for 'self-oriented' in all countries are closer to the upper limit of the scale (a ceiling effect), which suggests that the importance that universities attribute to instrumental rationales can hardly be increased because it is already very high. If there is still a difference of thinking in the central communication offices of the European university landscape, it is with regard to the importance attributed to motives such as enlightenment and public engagement.

## Predictors of communication activity

Our best fitting models are Models 3, as shown by the (highest) adjusted R squared (S5–S7 Tables). These models show that both context (C) and disposition (D) factors are important, explaining a significant amount of variance in universities' central offices communication activity. Models 1 show that the 'size' of the university and 'country' are not important for whether a university organises face-to-face events, that is, they do not depend on size of the university or location, but these variables make a difference for media communication both traditional and online. Larger universities often show increased media activity, and German universities tend to use less traditional and online means to communicate about their research than universities in Italy (the reference category) (see Fig 2).

Models 2 add resource-related variables, and show that having 'communications staff' and an institutional 'policy' for public communication are good predictors of increased face-to-face initiatives; these effects are kept significant in Models 3 when 'rationale variables' are introduced confirming their importance. Models 3, thus confirms that for public events, the most important factors are 'staff', 'policy' and a 'public-oriented' view of communication.

Similarly, for *traditional media channels*, Models 2 show the importance of 'policy and 'communications staff' for increased communication, yet, Models 3, with all variables considered, point to most important predictors being 'staff' and 'public-good' oriented rationales;

policy loses its importance. Similarly, in *new media channels*, it is only the 'public-good' oriented rationales that explain the increased use of new media in some universities, and the size of the university.

Overall, these findings suggest that universities engage more in media channels, while engagement in public events require more staff, a policy, and a 'public-good' oriented motivation, that is, increased commitment.

We note that the variable 'active researchers' is not a significant predictor of central communication activity, suggesting a subordinate role for individual researchers at this level. This contrasts with their important role at the meso level [25]. However, this seems plausible in medialised universities. In the media portrayal, the prominent media faces of a university (visible 'scientists') are often not its most prominent researchers, but people who are good communicators of others' research and are loved by the media for it; they are spontaneously available (fast), talk understandably (unscientifically), and are already known by the public. On the other hand, the activities of central offices are much more than science communication, so it also not a totally surprising finding.

## Discussion

In this paper, we provide an overview of the science communication function at universities in four European countries, Germany, Italy, Portugal, and the United Kingdom. The data points to transformations in the way central communication offices do science communication and provides insights about its role in the university landscape.

Central communication offices are established structures in most universities, with human and financial resources, strategies and policies for communicating in general and also for science. Yet, and despite the available infrastructure, the evidence clearly shows that science communication is a secondary function at this level, while suggesting that central offices play a supporting role to that activity.

First and foremost, only a part of the daily communication activities of central offices is about scientific research. Although communication teams are reasonably sized, the range of backgrounds and professional positions, often focused on media and digital communication and less in science communication, challenge the role of public engagement 'experts' [43]. In increasingly multi-functional offices, communication activities respond to the various university missions and priorities, and communicators become 'multiprofessionals' excelling in multi-tasking. With PR and marketing activities dominating central communication offices, the view also for communicating about scientific research is mostly aligned with self-interested priorities and the practice constrained by the skills sets available at central offices, which adopt a supporting role.

The data suggests that central offices are moving away from 'doing' science communication to facilitating and motivating it, throughout support activities, while focusing on university priorities of attracting students and monitoring educational policy [22]. This is a silent break with the long-practiced standard model of university science *communication. In this model, the financial and human resources for* communication were concentrated at the top of the organisation, and the division of labor between researchers and professional communicators was clearly distinguished [47, 48]. For example, it was common for researchers (voluntarily or after prompting) to feed communications offices with research material assuming that professionals would refine it into 'good' science communication and 'popularise' it through the right channels. This traditional practice seems to be on the decline, at least it is no longer the expected role of centralised communication of the university. Nowadays, central communication departments are adopting a supporting role in communicating science by facilitating the activity of scientists and research institutes [25] and disseminating science more broadly.

In our data, this was in evidence in the *infrastructure* available at central offices—little is allocated to science communication (neither staff nor funding), just enough to support the activities at other levels, while making use of the resources already available for other purposes—i.e. skilled staff and the university official channels (e.g. website and social media). As such the *scope of activities* performed at this level focuses on supporting scientists' media relations to disseminating news in the university channels, maintaining the website and social media, or supporting the organisation of university public events.

Moreover, among a spectrum of activities that universities use to communicate about their research, the focus is on media outlets, both traditional and online. On the contrary, face-to-face activities to communicate and discuss science are less performed at this level, which nevertheless offer public lectures, open days, and exhibitions; two-way formats for public discussion and debate were rarely reported. This indicates a priority for medialised communication of science and a disconnection between the universities' stated vision for public engagement and their actual investment. A focus on media activities and news values means that despite the ideals for public engagement and public debate, public engagement of any substantive kind is largely 'rhetorical' and real engagement activities are neglected and not the remit of central offices. Despite a recognition of a public-good motivation, in practice communication serves instrumental goals for broad visibility, potentially compromising the mission of universities in involving citizens in scientific research and contributing to the 'societal conversation of science'.

Communication of science as a routine activity in central offices appears then dispersed, multitasked, and disconnected from the activities of researchers. Researchers are more or less irrelevant for central communication activities that are taken up by professional communicators. The division of labour seems to be changing: it has now become common for researchers who want to see their research findings disseminated, to bring the results ready-prepared for text or visual media—for example, an entry for the university blog or a press release is now often done by the researcher on a funded project, which will have its own obligations for dissemination written into it; the central office will support the dissemination but will not initiate or conduct it alone. Entradas and co-authors (2023) show how this media activity has become a shared practice.

This facilitating rather than acting role is present in everyday observations of the institutional environment: the frequent appeals to motivate scientists to engage in science communication, the setting of incentives (monetary or, increasingly important, time resources such as release from teaching and administrative tasks) and through formal policies that make 'communication' a recruitment or promotion criteria. It is also seen in separate funding schemes and initiatives for science communication, prizes that are being awarded, and university training initiatives for researchers to communicate publicly (e.g. media a public speaking), continuing education programs, and science communication workshops [49, 50].

Overall, this process must be understood as *internal outsourcing*, as has become common in service and production companies of all sectors of industry since the 21st century. It reduces costs and improves efficiency of the central administration by focusing on core capabilities [51]. It is another indication of the adoption of corporate practices within publicly funded universities, which has been widely described in the literature as the New Public Management reforms in European higher education system [52]. The internal outsourcing of costly public engagement activities to subordinate units is intended to release management capacity at the top level, for which functions such as marketing, public affairs and public relations are becoming increasingly the focus. In line with the dictum to *'do what you can do best and outsource the rest'*, the gradual withdrawal of central offices from doing science communication can also be rationalised; it sounds plausible that science communication should be done where the

actual action takes place, while 'selling' the university as a whole is best managed from central structures that build the university's public image. However, resources often remain at the central level; lack of resources locally might then be a 'boundary-block' that undermines public engagement practice in reality [43: 1299].

Finally, the similar trends in science communication practice across countries point to similar challenges in universities. Institutions compete for the attention of the same audiences and use the same means to reach their goals. Yet, there are also notable differences that give rise to functionally equivalent diversity rather than standardisation of communication practices in European universities. In Germany and British universities we find a stronger focus on self-oriented rationales and a significantly lower recognition of public-good oriented rationales. This indicates a larger disconnect between the university vision for public engagement and its actual practice. This is intriguing. These countries have pioneered the 'Public Understanding of Science' movement [53], particularly the UK, have significantly larger communications teams pushing other communication functions; This seems above all an important sign of stronger marketisation of university education and commercialisation of science in these countries.

## Conclusion

This study pioneers the systematic mapping of science communication practices at central communication offices of the modern university. Using robust data and a systematic methodology allowed us to distinguish practices at the central from other organisational levels, and science communication from other communication functions. We characterised science communication at the central level in activities, resources (staff, funding and policies), audiences, rationales (public-good oriented, self-oriented), and drivers of activities. The evidence shows that science communication is a secondary function at the central offices, used as an instrument for gaining institutional visibility. At the same time, this points to a redefined role for central offices: they move away from doing it while moving towards facilitating it; thus outsourcing science communication to researchers (individual level) and to institutes and departments (meso-level), while keeping their focus on institutional PR and related core capabilities.

This diminished space for science communication at central structures of the university might compromise universities' capacity to practice their official mission of public engagement by limiting the development of communication formats that align with values and long-term goals of public engagement. Nevertheless, this facilitating role at the centre is indicative of building an infrastructure that promotes science communication by sharing resources and skills. How this works out and what this means in practice remains to be seen; it seems to be work in progress at modern universities in Europe and beyond.

## Supporting information

**S1 Table. Descriptives for indices from sum of activities.**
(XLSX)

**S2 Table.** (a) Rationales for communication (n = 319). (b) Table. Exploratory factor analysis loadings for rationales (n = 319).
(ZIP)

**S3 Table. One-way ANOVA.** Analysis of variance for communication activities. Abreviations: Sum of squares (SS); df (degrees of freedom); MS (Mean Square); F statistic (F) and (p (significance value).
(XLSX)

**S4 Table. Analysis of variance for rationales.** Abbreviations: Sum of squares (SS); df (degrees of freedom); MS (Mean Square); F statistic (F) and (p (significance value).
(XLSX)

**S5 Table. Summary of hierarchical regression analysis for institutional public communication activity—public events (n = 220).**
(XLSX)

**S6 Table. Summary of hierarchical regression analysis for institutional public communication activity—traditional media channels (n = 220).**
(XLSX)

**S7 Table. Summary of hierarchical regression analysis for institutional public communication activity—new media channels (N = 220).**
(XLSX)

**S1 File.**
(PDF)

## Author Contributions

**Conceptualization:** Marta Entradas, Frank Marcinkowski, Martin W. Bauer, Giuseppe Pellegrini.

**Data curation:** Marta Entradas, Frank Marcinkowski, Martin W. Bauer, Giuseppe Pellegrini.

**Formal analysis:** Marta Entradas.

**Funding acquisition:** Marta Entradas.

**Investigation:** Marta Entradas, Frank Marcinkowski, Martin W. Bauer, Giuseppe Pellegrini.

**Methodology:** Marta Entradas, Martin W. Bauer.

**Project administration:** Marta Entradas.

**Supervision:** Marta Entradas.

**Writing – original draft:** Marta Entradas.

**Writing – review & editing:** Marta Entradas, Frank Marcinkowski, Martin W. Bauer, Giuseppe Pellegrini.

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
