## [Decision Letter · Decision Letter 0]

6 Jul 2023

PONE-D-23-18560University central communication offices moving away from doing towards facilitating science communication at other levelsPLOS ONE

Dear Dr. Entradas,

Thank you for submitting your manuscript to PLOS ONE. After careful consideration, we feel that it has merit but does not fully meet PLOS ONE’s publication criteria as it currently stands. Therefore, we invite you to submit a revised version of the manuscript that addresses the points raised during the review process.

We look forward to receiving your revised manuscript.

Kind regards,

Radoslaw Wolniak, full professor

Academic Editor

PLOS ONE

Journal Requirements:

   "This research received funding from Fundação para a Ciência e Tecnologia (FCT), with Grant agreement No. PTDC/COM-OUT/30022/2017."

Reviewers' comments:

Reviewer's Responses to Questions

**Comments to the Author**

1. Is the manuscript technically sound, and do the data support the conclusions?

Reviewer #1: Yes

Reviewer #2: Yes

2. Has the statistical analysis been performed appropriately and rigorously? 

Reviewer #1: Yes

Reviewer #2: Yes

3. Have the authors made all data underlying the findings in their manuscript fully available?

Reviewer #1: Yes

Reviewer #2: Yes

4. Is the manuscript presented in an intelligible fashion and written in standard English?

Reviewer #1: Yes

Reviewer #2: Yes

5. Review Comments to the Author

Reviewer #1: Dear Authors,

Congratulations for your interesting research.

The title and the abstract coincide with the content of the paper. Keywords are well-chosen.

• The problem and goal of the paper are clearly and correctly formulated.

• The manuscript is clear, relevant for the field and presented in a well-structured manner.

• To achieve the goal of the paper, the applied research methods can be considered correct.

• The paper contains a comprehensive reference list. The literature studies presented should be considered sufficient both as to the proper selection of sources and their quantity. The cited references are current.

• The manuscript’s results are reproducible based on the details given in the methods section.

• The figures and the tables are appropriate. They properly show the data. They are easy to interpret and understand.

I have some suggestions on how to make your text more attractive

In conclusion, I propose

- highlight the new knowledge and the lessons learned from it,

- describe the importance of the research and how it affects the wider field, show how the information obtained can be further used

Conclusions must be clearly and unambiguously linked to the results of the survey. Their theoretical and practical implications should be indicated

Reviewer #2: Article at a very high level of content. A relatively broad review of the literature underpins the considerations. The research tools used are properly presented and the analysis of the results obtained correctly interpreted. The discussion is also an obvious added value of this article.

6. PLOS authors have the option to publish the peer review history of their article (what does this mean?). If published, this will include your full peer review and any attached files.

Reviewer #1: No

Reviewer #2: No

---

## [Author Response · Author response to Decision Letter 0]

2 Aug 2023

Dear PloOne Editor in Chief,

I am submitting the revised manuscript “University central offices are moving away from doing towards facilitating science communication: a European cross-comparison”; we tweaked slightly the title of the manuscript to better reflect the scope of the study.

We appreciated the reviewers’ comments, which we understood having very little to be changed or added. Reviewer 2 had no suggestions for changes, and reviewer 1 suggested to highlight the findings of this important study, which we did in the conclusions of the paper.

We are also submitting the ethical approval declaration for the study, and all procedures of participant involvement have been explained in the methods section of the manuscript.

---

## [Editor Report · Decision Letter 1]

10 Aug 2023

University central offices are moving away from doing towards facilitating science communication: a European cross-comparison

PONE-D-23-18560R1

Dear Dr. Entradas,

We’re pleased to inform you that your manuscript has been judged scientifically suitable for publication and will be formally accepted for publication once it meets all outstanding technical requirements.

Kind regards,

Radoslaw Wolniak, full professor

Academic Editor

PLOS ONE
---

## [Editor Report · Acceptance letter]

7 Sep 2023

PONE-D-23-18560R1 

University central offices are moving away from doing towards facilitating science communication: a European cross-comparison 

Dear Dr. Entradas:

I'm pleased to inform you that your manuscript has been deemed suitable for publication in PLOS ONE. Congratulations! Your manuscript is now with our production department. 

Kind regards, 

on behalf of

Professor Radoslaw Wolniak 

Academic Editor

PLOS ONE